# Discontinuation of Imatinib in Children with Chronic Myeloid Leukemia: A Study from the International Registry of Childhood CML

**DOI:** 10.3390/cancers13164102

**Published:** 2021-08-15

**Authors:** Frédéric Millot, Meinolf Suttorp, Stéphanie Ragot, Guy Leverger, Jean-Hugues Dalle, Caroline Thomas, Nathalie Cheikh, Brigitte Nelken, Marilyne Poirée, Geneviève Plat, Birgitta Versluys, Birgitte Lausen, Marina Borisevich

**Affiliations:** 1Inserm CIC 1402, University Hospital of Poitiers, 86000 Poitiers, France; stephanie.ragot@univ-poitiers.fr; 2Medical Faculty, Pediatric Hemato-Oncology, Technical University Dresden, 01307 Dresden, Germany; meinolf.Suttorp@uniklinikum-dresden.de; 3Department of Pediatric Hematology, Trousseau Hospital, 75012 Paris, France; guy.leverger@aphp.fr; 4Department of Pediatric Hematology, Robert Debré University Hospital, 75019 Paris, France; jean-hugues.dalle@aphp.fr; 5Department of Pediatric Hematology, University Hospital of Nantes, 44000 Nantes, France; caroline.thomas@chu-nantes.fr; 6Department of Pediatric Hematology, University Hospital of Besançon, 25056 Besançon, France; ncheikh@chu-besancon.fr; 7Department of Pediatric Hematology, University Hospital of Lille, 59000 Lille, France; brigitte.nelken@chru-lille.fr; 8Department of Pediatric Hematology, University Hospital of Nice, 06000 Nice, France; poiree.m@chu-nice.fr; 9Department of Pediatric Hematology, University Hospital of Toulouse, 31000 Toulouse, France; plat.g@chu-toulouse.fr; 10Department of Hematology, University Medical Center Utrecht, 3584 Utrecht, The Netherlands; A.B.Versluijs@prinsesmaximacentrum.nl; 11Department of Pediatrics, Rigshospitalet, 2100 Copenhagen, Denmark; Birgitte.Lausen@regionh.dk; 12Belarusian Research Centre for Pediatric Oncology, Hematology and Immunology, 223053 Minsk, Belarus; borisevich10@mail.ru

**Keywords:** chronic myeloid leukemia, imatinib, children

## Abstract

**Simple Summary:**

About 50% of adults with chronic myeloid leukemia (CML) in sustained deep molecular response (DMR) to tyrosine kinase inhibitors (TKI) could discontinue the treatment permanently without molecular relapse. Recommendations regarding discontinuation apply only for adults because childhood CML is a very rare disease and represents a separate entity. The aim of our retrospective study was to assess within the International Registry of Childhood CML, the rate of children remaining in molecular response after discontinuation of imatinib in a context of DMR defined as BCR-ABL1/ABL1 < 0.01% (MR^4^) for at least two years. Eighteen patients less than 18 years old at diagnosis of CML exhibiting a sustained DMR followed by imatinib discontinuation were identified. After discontinuation, the molecular free remission rate was 61%, 56% and 56% at 6, 12 and 36 months, respectively. Our findings represent the basis of recommendation regarding discontinuation for physicians involved in the pediatric CML field.

**Abstract:**

Within the International Registry of Childhood Chronic Myeloid Leukemia (CML), we identified 18 patients less than 18 years old at diagnosis of CML who were in the chronic phase and exhibiting a sustained deep molecular response (DMR) to imatinib defined as BCR-ABL1/ABL1 < 0.01% (MR^4^) for at least two years followed by discontinuation of imatinib. Before discontinuation, the median duration of imatinib was 73.2 months (range, 32–109) and the median duration of MR^4^ was 46.2 months (range, 23.9–98.6). Seven patients experienced loss of major molecular response (MMR) 4.1 months (range, 1.9–6.4) after stopping and so restarted imatinib. The median molecular follow-up after discontinuation was 51 months (range, 6–100) for the nine patients without molecular relapse. The molecular free remission rate was 61% (95% CI, 38–83%), 56% (95% CI, 33–79%) and 56% (95% CI, 33–79%) at 6, 12 and 36 months, respectively. Six of the seven children who experienced molecular relapse after discontinuation regained DMR (median, 4.7 months; range, 2.5–18) after restarting imatinib. No withdrawal syndrome was observed. In univariate analysis, age, sex, Sokal and ELTS scores, imatinib treatment and DMR durations before discontinuation had no influence on treatment free remission. These data suggest that imatinib can be safely discontinued in children with sustained MR^4^ for at least two years.

## 1. Introduction

Chronic myeloid leukemia (CML) is characterized by the Philadelphia chromosome with the BCR-ABL1 fusion gene coding for a protein tyrosine kinase [1,2]. The identification of this protein as a necessary and sufficient driver of the leukemia process led to the development of BCR-ABL1 tyrosine kinase inhibitors (TKIs) [3]. The prognosis of patients with CML has considerably improved over the last 20 years with the introduction of TKIs to the treatment strategy. Prospective studies demonstrated that imatinib, a TKI of the first generation, is efficient as first-line therapy in children with CML in the chronic phase (CP) [4,5]. Lifelong treatment for children treated with TKI during their period of growth and development may expose them to unique side effects [6]. For these reasons, the feasibility of TKI discontinuation in children with a sufficient TKI response must be explored. Studies in adults with CML demonstrated that about 50% of patients with a deep and sustained molecular response to imatinib, nilotinib or dasatinib could discontinue TKI permanently without molecular relapse [7,8,9,10,11]. Thus, treatment free remission (TFR) may be the main goal for patient with CML according to the recent European LeukemiaNet (ELN) recommendations [12]. The National Comprehensive Cancer Network (NCCN) guidelines and ELN recommendations propose discontinuation of TKI in selected adults after prolonged treatment with TKIs and maintenance of stable deep molecular response for several years [12,13]. However, these recommendations apply only for adult patients because childhood CML represents a separate entity [14,15]. Data regarding TKI discontinuation in children with CML are reported for a limited number of patients [16,17,18]. The International Registry of Childhood Chronic Myeloid Leukemia provided a valuable collection of data to assess the discontinuation of imatinib in a cohort of children in MR^4^ for two years or more [19].

## 2. Materials and Methods

The International Registry of Childhood Chronic Myeloid Leukemia (I-CML-Ped Study registered at www.clinicaltrials.gov NCT01281735) is enrolling patients less than 18 years of age at the diagnosis of CML in CP or advanced phase according to the criteria of the European LeukemiaNet (ELN) [20]. The I-CML-Ped Study was set up in order to describe the disease and to assess the management and outcome of CML in children and adolescents. Data regarding sex, age, clinical and biological characteristics and lines of treatment were collected. The I-CML-Ped Study was not designed to collect data related to a TKI withdrawal syndrome characterized by the occurrence of musculoskeletal pain after TKI discontinuation. Investigators were retrospectively solicited to check their patients’ files in order to seek for this information. Enrollment was done retrospectively from January 2000 until December 2010 and then prospectively from January 2011. The I-CML-Ped Study was approved by the institutional review committee of the University Hospital of Poitiers (Poitiers, France) in accordance with the Declaration of Helsinki. Written informed consent was obtained from the children and/or their legal guardians.

Determination of the Sokal score was performed according to the mathematical equation for patients aged less than 45 years [21]. Calculation for the EUTOS long-term survival (ELTS) score was determined as previously reported [22].

Analysis of BCR-ABL1 transcript levels in the peripheral blood (PB) and bone marrow was performed by local laboratories using quantitative reverse transcriptase-polymerase chain reaction (RT-qPCR), and the results were depicted according to the International Scale (IS) [23]. A major molecular response (MMR) was defined as a ratio of BCR-ABL1/ABL1 less than 0.1% and a deep molecular response (DMR) as a ratio of BCR-ABL1/ABL1 of 0.01% (MR^4^) or lower. We retrospectively selected children in CP at diagnosis treated with imatinib front line without failure or suboptimal response, or who switched to another TKI before discontinuation, and who exhibited continuous DMR for at least two years and then discontinued the TKI. The suboptimal response and failure of treatment were defined according to the ELN criteria [20]. The outcomes of these children and TFR rates at defined time points were retrospectively analyzed. BCR-ABL1 transcripts were monitored in the peripheral blood by RT-qPCR after TKI cessation. Molecular relapse was defined as a loss of MMR at any time. Molecular relapse-free survival and survival in TFR from imatinib discontinuation was estimated by the Kaplan-Meier method. Comparisons of age, sex, Sokal (less than 45 years old) and ELTS scores, imatinib treatment duration before discontinuation and duration of DMR until imatinib discontinuation between children with relapse and children without relapse were performed using a chi-square test or Fisher’s exact test for qualitative variables and using the Wilcoxon signed-rank test for quantitative variables.

## 3. Results

Between January 2000 and June 2020, there were 581 patients from 16 countries with a reported diagnosis of CML enrolled in the I-CML-Ped Study. Among them, 495 patients were in CP and 86 in advanced phase at diagnosis according to the ELN criteria. We retrospectively identified 18 children with CML in CP at diagnosis and with sustained DMR for at least two years who discontinued imatinib as an attempt to achieve TFR in the years 2008 to 2018. During the same period of time, 30 children with CML in CP met the same criteria for TKI stopping without being attempted. However, the reason was not specified in the data collected. All patients in sustained MR^4^ for at least two years who did or did not discontinue imatinib represent 9% of the total cohort of patients diagnosed in CP. The characteristics of the 18 patients who discontinued imatinib are reported in Table 1. 

The median dose of imatinib administered at the start of the treatment was 260 mg/m^2^ (range, 204–400). From diagnosis of CML until imatinib discontinuation, all 18 children and adolescents showed no progression, resistance, a suboptimal response or were switched to another TKI. The median time to achieve MMR and DMR was 6.4 months (range, 1.9–19.2) and 12 months (range, 3–50) after the start of imatinib, respectively. The median treatment duration of imatinib was 73 months (range, 32–109) before discontinuation. The median age at discontinuation of TKI was 16 years (range, 9–24). Prior to discontinuation, the median duration of sustained DMR was 46.2 months (range, 24–98.6) and the median number of successive RT-qPCR measurements less or equal to MR^4^ was nine (range, 3–21) during this period of time. All 18 patients discontinued imatinib abruptly. In all children excepted one, the cessation was decided in accordance with the referring physician. The remaining patient, who was 16 years old, decided by himself to stop imatinib without informing his physician and without changing the rhythm of the monitoring of the transcript level. Except for this patient, the transcript level was monitored on a monthly basis during the first year after cessation and extended from two to six months thereafter. The course of each patient is reported in Figure 1. 

Among the 18 children, imatinib was resumed in nine (same dosage as before discontinuation in six patients, unknown in two and increased dose in one) from 2.2 to 9.2 months (median: 4.4) after discontinuation of imatinib. Seven of these nine children experienced a molecular relapse with a median time of 4.1 months (range, 1.9–6.4) after stopping, leading to imatinib resumption within a median period of 35 days (range, 7–148) after MMR loss. The delay of imatinib resumption is explained by the decision of the clinician to perform additional determinations of the transcript level to confirm the loss of MMR in two children with an increase in the BCR-ABL1/ABL1 ratio slightly above 0.1%. Depending upon the decision of the physician, the remaining two patients resumed imatinib 3.6 and 3.4 months after discontinuation because of a one-log fold increase in transcript level (ratio of BCR-ABL1/ABL1 increased from 0.001% to 0.01% and 0.012%, respectively) but without any loss of MMR. The proportion of patients maintaining molecular free remission was 61% (95% CI, 38–83%), 56% (95% CI, 33–79%) and 56% (95% CI, 33–79%) at 6, 12 and 36 months, respectively (Figure 2). 

The median molecular follow-up after discontinuation was 51 months (range, 6–100 months) for these seven patients in molecular free remission.

Six of the seven children who experienced molecular relapse after discontinuation reachieved MR^4^ at a median time of 4.7 months (range, 2.5–18) after restarting imatinib; the remaining patient achieved MMR but not DMR. 

In univariate analysis, age, sex, Sokal (less than 45 years old) and ELTS scores, imatinib treatment duration before discontinuation and duration of DMR until imatinib discontinuation had no influence on treatment free remission success.

So far, two children have experienced a second attempt at discontinuation. The durations of MR^4^ before the first attempt were 2 and 5.1 years, respectively, and the time to loss of MMR after this first discontinuation was two and four months with a regain of MR^4^ within 5.3 and 18.1 months. A second discontinuation was performed in these two patients after a MR^4^ duration of 21 months and 4 years, respectively. This resulted in a loss of MMR three months after discontinuation in the first patient, and in the second patient, in a loss of MR^4^ leading to the resumption of imatinib six months after discontinuation. The first patient regained MR^4^ within two months and was in persistent MR^4^ eight months after resumption, whilst the second patient regained MMR within nine months and was in persistent MMR 40 months after resumption, however, in a context of poor compliance.

With a median follow-up from diagnosis of 107 months (range, 67–209), all the 18 children are alive without any case of progression to an advanced phase.

## 4. Discussion

We found that discontinuation of imatinib in children with CML in a deep and sustained molecular response resulted in a molecular free survival rate of 56% (95% CI, 33–79%). The present study indicated that the proportion of molecular relapse after discontinuation is similar to the results reported in cohorts of adults and in a prospective study conducted in children in Japan with a TFR rate of approximately 50% [10,18,24]. Studies in adults reported the possibility of cessation of second-generation TKIs such as nilotinib and dasatinib but there are no available data regarding the pediatric population [9,25]. The present findings contrast with the reported lower proportion (23%) of children maintaining a treatment free remission in the Stop Imaped study [17]. However, this difference can be easily explained by a less stringent criterion of molecular relapse (loss of MMR in the present study) than in the Stop Imaped study (loss of MR^4^ at any time point). The definition of molecular recurrence has a direct impact on the rate of TFR [11]. Since the first reports of successfully stopping TKI treatment, loss of MMR has become the current consensual definition of molecular relapse in a context of discontinuation of TKI [12,13].

The timing of relapse (mainly during the first six months after discontinuation) observed in the present pediatric cohort is similar to the kinetic of relapse reported in cohorts of adults in which molecular losses occurred in 82% and 95% of the cases within the first 6 and 12 months, respectively [11].

In the International Registry for Childhood CML, 48 (9%) patients achieved MR^4^ and maintained this low disease burden for at least two years, and thus would be potential candidates to discontinue the treatment. However, a notable proportion of children, although fulfilling the criteria for TKI stoppage, did not undertake this attempt. Fear and anxiety of the treating physician, the parents or the patient could be possible underlying reasons as well as a lack of clear guidelines related to TKI cessation in children. On the back of the encouraging data presented here, it might be expected that more patients will take the opportunity to attempt to stop their TKI in the future.

The lack of published recommendations in children with CML at the time of TKI stoppage could explain why the treatment was re-initiated in two of our children of the cohort described here showing fluctuations of the transcript levels during cessation while still remaining in MMR. In adults, 2% of the patients enrolled into the Euro-Ski study restarted TKI before loss of MMR was observed [10]. 

The criterion of maintaining MR^4^ for at least two years prior to an attempt at discontinuation in our patients approaches the criteria of the NCCN guidelines (minimal duration of TKI treatment three years with at least two years of having achieved MR^4^) more so than the ELN recommendations based on at least five years of treatment with at least two years in MR^4.5^ [12,13].

Relapsing patients must resume treatment. All pediatric patients reported here experiencing a loss of MMR after TKI discontinuation were sensitive to retreatment with imatinib and regained MMR. Also, in adult cohorts, a molecular response is usually regained after resuming the initial treatment or after a switch to a second generation in case of insufficient response [11,26].

A withdrawal syndrome characterized by the occurrence of musculoskeletal pain in adults after stopping imatinib therapy was not reported in the present cohort of 18 children [27,28,29]. However, because of the retrospective nature of the present study, we cannot exclude underreporting of mild musculoskeletal symptoms. Nevertheless, our data are in line with the retrospective Japanese Stop TKI trial, which also observed no withdrawal symptoms in the pediatric cohort [18]. Keeping in mind the so far reported small number of patients at a minor age undertaking a stoppage attempt, the frequency of the occurrence of a withdrawal syndrome still remains to be determined in the pediatric population. 

Several factors are associated with a successful TFR after TKI discontinuation in adult cohorts [4]. The most commonly reported factors are a prolonged duration of TKI treatment and DMR before discontinuation and the depth of molecular response [10]. A lower CML risk score was also reported as predictive of maintaining treatment free remission after discontinuation [24]. Initial de-escalation of imatinib before cessation as performed in the DESTINY trial rather than abrupt discontinuation might improve the TFR success [30]. The optimal modalities of discontinuation (gradual de-escalation, low TKI doses, intermittent TKI administration) are currently being investigated in adults with CML. We did not find any influence of age, sex, Sokal (less than 45 years old) and ELTS scores, imatinib treatment duration before discontinuation and duration of DMR until imatinib discontinuation on TFR success in the present study. The limited size of this cohort of pediatric patients can explain the limited significance of this univariate analysis of predictive factors for successful TFR. Predictive factors of TFR success were not also identified in the Japanese cohort of children [18]. Thus, such factors still remain to be determined in pediatric patients with CML.

With a median follow-up from diagnosis of 107 months (range, 67–209), all the 18 children are alive without any case of progression to the advanced phase, suggesting that discontinuing TKIs had no negative impact on clinical outcomes. Occurrence of progression of CML to a blastic phase remains exceptional in adults in a context of failure of TFR [8].

It was reported in adults with CML that a failure of a first attempt of discontinuation does not preclude a second successful discontinuation [31]. Two of our patients experienced an unsuccessful attempt at discontinuation of imatinib after achievement of a second period of deep and sustained molecular response. An early molecular relapse within the first three months after the first discontinuation attempt was recently reported as a factor significantly associated with the failure at a second attempt [32].

## 5. Conclusions

This study indicates that imatinib could be discontinued without an impact on outcomes in patients younger than 18 years of age at diagnosis of CML who responded well to the treatment (no failure, no suboptimal response, no switch to another TKI) and achieved sustained MR^4^ for at least two years with imatinib. Our findings represent the basis of recommendations for physicians involved in the pediatric CML field. Larger studies of TKI discontinuation in children with CML are needed in order to identify factors predicting TFR success.

## Figures and Tables

**Figure 1 cancers-13-04102-f001:**
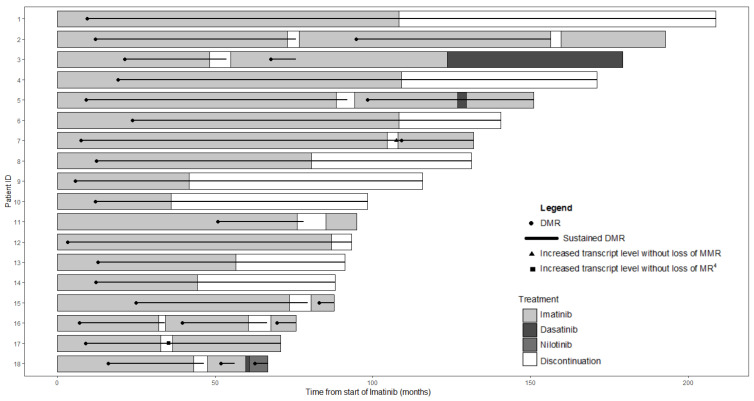
Swimmer plot depicting treatment course of each patient according to the time from the start of imatinib. Each band represents an individual patient. The bands are color-coded based on the patients’ treatment. In each band, a black line indicates periods of sustained deep molecule response (DMR). Patients 3 and 18 were switched to second-generation tyrosine kinase inhibitors because of unstable DMR after imatinib resumption. Dasatinib was introduced in patient 5 in order to optimize the DMR but was quickly stopped because of side effects. MMR, major molecular response; MR^4^, ratio BCR-ABL1/ABL1 of 0.01%.

**Figure 2 cancers-13-04102-f002:**
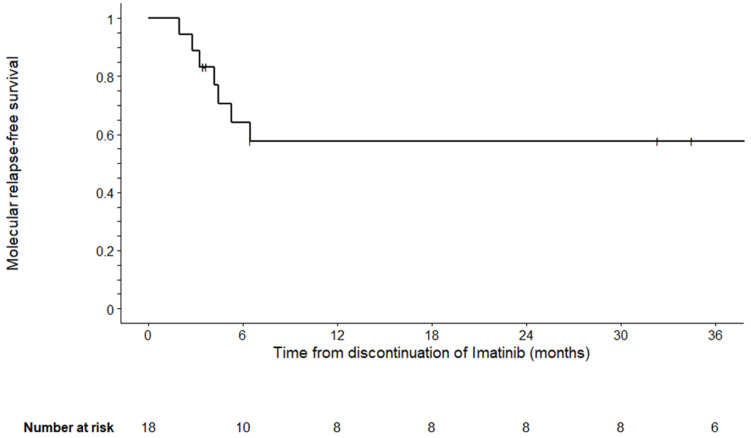
Molecular relapse-free survival after imatinib discontinuation (*N* = 18).

**Table 1 cancers-13-04102-t001:** Patients’ characteristics.

**Characteristics**	*N* = 18
**Sex, N**	
Boys	11
Girls	7
**Median age at diagnosis of CML, years**	11.9 (range, 2.3–15.8)
**Median white blood cell count at diagnosis, 10^−9^/L**	77.8 (19.2–352.7)
**Transcript type**	
b3a2	10
b2a2	1
b3a2 and b2a2	3
Unknown	4
**Sokal risk score (<45 years), N**	
Low	6
Intermediate	4
High	5
Missing	3
**ELTS score, N**	
Low	12
Intermediate	2
High	2
Missing	2

Abbreviations: CML, chronic myeloid leukemia; ELTS, Eutos long-term survival; N, number of patients.

## Data Availability

The data that support the findings of this study are available on request from the corresponding author. The data are not publicly available because of privacy issues.

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
