# Peer review of "Discontinuation of Imatinib in Children with Chronic Myeloid Leukemia: A Study from the International Registry of Childhood CML"

_cancers, 2021, doi:10.3390/cancers13164102_

Round 1

Reviewer 1 Report

The Cancers manuscript 1308083 entitled „Discontinuation of Imatinib in Children with Chronic Myeloid 2 Leukemia” by Frédéric Millot and colleagues summarizes the experience of the International Registry of Childhood CML with discontinuation of imatinib in pediatric CML patients. Although the number of patients is small, the results are very important and very encouraging. The study gives clear direction, when to consider imatinib cessation in pediatric CML patients and how to monitor the patients afterwards.

Author Response

The authors thank the reviewer 1 for the encouraging comments regarding this work 

Reviewer 2 Report

The authors report the DMR at least 2 years after discontinuation of imatinib in a children cohort from the International Registry of CML. Based on the method section, 18 children achieved continuous DMR at least 2 years of initial treatment of imatinib are selected for analysing. The median treatment duration of imatinib was 73 months (range 32-109). Accordingly, the major finding shows the molecular free remission rate at 0.5, 1 and 3-yr is 61%, 56% and 56%, respectively.

Here are some comments to scrutinize and understand this study for better reading.

1. Inadequate presentation of result:
Because the intent of this study wanted to address the rate of >= DMR at 2-yrs of discontinuation of imatinib, the context and figure did not show the 2-yr molecular free rate at 2-yr.

2. Avoiding misunderstandings:
Enrolled number of subjects is small in this manuscript, to show time dependent clinical course in detail by a swimmer's plot would be better.

3. Which result here is analyzed by Chi-square test or Fisher's exact test and Wilcoxon signed-rank test.

4. The first sentence of Abstract is plagiarized from Simple Summary.

5. Several typo errors:
#line 4: CML. --> "." is not necessary
#line 7: Borisevich12. --> "." is not necessary
#line 26-27: deep and sustained molecular response (DMR) --> sustained deep ... (DMR)
#line 37: (CML) is redundant
#line 59: Prospectives --> Prospective
#line 67: ELN --> European LeukemiaNet (ELN)
#line 68: European LeukemiaNet (ELN) --> ELN
#line 129: 1.94 - 19.25 --> " -" replace by "-", and the all numbers in this paper are almost "one decimal place"
#line 143: 1.9-6.4 months --> 1.9-6.4; "months" is redundant
#line 178, 179: 33%-79 --> 33%-79%)

6. Figure resolution is too low.

7. The format of "References 1 and 2" is discordant.

Author Response

Please find attached our responses 
